# Genetic Gain and Inbreeding in Different Simulated Genomic Selection Schemes for Grain Yield and Oil Content in Safflower

**DOI:** 10.3390/plants13111577

**Published:** 2024-06-06

**Authors:** Huanhuan Zhao, Majid Khansefid, Zibei Lin, Matthew J. Hayden

**Affiliations:** 1School of Applied Systems Biology, La Trobe University, Bundoora, VIC 3083, Australia; majid.khansefid@agriculture.vic.gov.au; 2Agriculture Victoria, AgriBio, Centre for AgriBioscience, Bundoora, VIC 3083, Australia; zibei.lin@agriculture.vic.gov.au

**Keywords:** genetic gain, inbreeding coefficient, safflower, simulation, grain yield, seed oil content

## Abstract

Safflower (*Carthamus tinctorius* L.) is a multipurpose minor crop consumed by developed and developing nations around the world with limited research funding and genetic resources. Genomic selection (GS) is an effective modern breeding tool that can help to fast-track the genetic diversity preserved in genebank collections to facilitate rapid and efficient germplasm improvement and variety development. In the present study, we simulated four GS strategies to compare genetic gains and inbreeding during breeding cycles in a safflower recurrent selection breeding program targeting grain yield (GY) and seed oil content (OL). We observed positive genetic gains over cycles in all four GS strategies, where the first cycle delivered the largest genetic gain. Single-trait GS strategies had the greatest gain for the target trait but had very limited genetic improvement for the other trait. Simultaneous selection for GY and OL via indices indicated higher gains for both traits than crossing between the two single-trait independent culling strategies. The multi-trait GS strategy with mating relationship control (GS_GY + OL + Rel) resulted in a lower inbreeding coefficeint but a similar gain compared to that of the GS_GY + OL (without inbreeding control) strategy after a few cycles. Our findings lay the foundation for future safflower GS breeding.

## 1. Introduction

Genomic selection (GS) is a modern breeding tool that uses genome-wide molecular marker information to predict the genomic estimated breeding values (GEBVs) of selection candidates (test individuals) to facilitate selection. To perform GS, a training population (TP), which has been genotyped and phenotyped, is used to predict the performance of the test individuals, which has been genotyped but not phenotyped by a statistical model [1,2]. 

GS has been applied in animal breeding and has resulted in increased genetic gain (ΔG) in the dairy cattle, beef cattle, pig, and poultry industries [3]. In plant breeding, GS has been increasingly incorporated into breeding programs to increase genetic gain [4]. Compared to phenotypic selection (PS) or marker-assisted selection (MAS), GS could achieve higher genetic gain by imposing higher selection intensity, achieving higher selection accuracy, especially for difficult or expensive traits to measure and low heritable traits, and shortening breeding cycles [5]. GS resulted in an extra 10–20% genetic gain over PS for drought tolerance in a maize study [6]. By using simulation, Lin et al. (2016) [7] showed that applying GS could double to triple the genetic gain by a 4-year reduction in cycle time when incorporated into a ryegrass breeding program. To simultaneously select multiple traits, GS with the selection index method was investigated and the choice of the selection index strongly affects genetic gain for target traits [8]. Rapp et al. (2018) [9] reported that the efficiency of selection largely depends on the weight of the trait in the index when simultaneously improving grain yield and protein content in durum wheat.

The increased selection intensity and higher selection accuracy of GS lead to greater short-term gains from selection; however, GS may reduce long-term gains by decreasing genetic variation and increasing the rate of inbreeding (ΔF) [10,11]. Restricting relationships between selected parents to control mating candidates (mating design), selecting for favorable minor alleles or putting weight on within-family information in breeding value estimation have been examined to control the inbreeding [12,13]. Allier et al. showed that considering within-family variance was more efficient than optimal crossing selection in converting genetic diversity into short- and long-term genetic gains in a simulated recurrent breeding program [14]. Giving extra weight to the favorable alleles with low frequency could increase long-term genetic gain by up to 30.8% compared to unweighted methods [15]. Using a simulated perennial ryegrass breeding scheme, Lin et al. (2017) [16] compared three types of inbreeding control strategies, and the use of a simultaneously adjusted selection and mate allocation method was able to reduce inbreeding to one-third of the original genomic selection scheme.

In GS, phenotyping calibrates prediction models instead of serving selection, which profoundly impacts the breeding program structure. The scenarios in which GS is implemented into breeding programs to allocate breeding resources efficiently and maximize genetic gain have been discussed [17,18]. Computer simulation has been used in plant breeding as a cost-effective tool to allow researchers to explore different scenarios without conducting expensive field trials or laboratory experiments. It is also a valuable tool for efficiently allocating resources and comparing breeding schemes, especially for finding the optimal strategies to implement GS [19,20]. In wheat, the comparison of classical two-stage PS with three GS breeding strategies for a fixed budget showed that the use of GS was the most advantageous, especially when low GS prediction accuracy (0.3) was tested for grain yield [21]. Lorenz (2013) [22] reported that the prediction accuracy (PA) in resource allocation strategies differed between GS models when they simulated a single biparental double haploid (DH) population to study the impact of resource allocation decisions, such as population size and replications on GS. The number of parents, number of hybrids, tester update, and genomic prediction of hybrids were simulated in a hybrid rice breeding program, and the results indicated that genomic prediction of hybrid performance was feasible and that the largest breeding size tested had the highest genetic gain [23]. Several simulation software packages have been developed to facilitate the investigation of different strategies for implementing GS in breeding programs [24,25,26].

Fundamental genetic research and genetic improvement in safflower have been achieved by using conventional PS and MAS in safflower breeding [27]. However, efforts to achieve further genetic improvement in safflower have decreased due to the limited market, small budget, competition with other oil seed crops, etc. [28]. Given the increasing demand for biofuel and healthy edible oil, safflower breeding has regained interest in recent years [29]. It is important to implement GS in breeding programs to take advantage of this modern breeding tool to breed safflower efficiently and rapidly. In this study, we simulated a safflower recurrent breeding program targeting grain yield (GY) and seed oil content (OL) by directly exploiting the genetic diversity of a genebank collection via four GS strategies. The aims were twofold: (1) to simulate different GS strategies in a safflower recurrent breeding program targeting GY and OL; (2) to compare genetic gains and genetic diversity losses during each breeding cycle to provide practical knowledge for the simultaneous improvement of these two traits in future GS breeding efforts.

## 2. Materials and Methods

### 2.1. Simulation Outline

We simulated a safflower breeding program with a recurrent selection scheme. The breeding cycle is illustrated in Figure 1. We applied GS to compare the genetic gains of four GS implementation strategies (single-trait: GS_GY, GS_OL; multi-trait: GS_GY + OL; multi-trait + inbreeding control: GS_GY + OL + Rel). The breeding cycle started with randomly selecting 50 individuals out of the 349 diverse safflower accessions and breeding lines as the initial crossing parents for cycle 1. A total of 30 biparental crosses were made within the 50 individuals by random crossing. We kept 10 F_1_ individuals per cross and advanced them to F_2_, with 200 F_2_ plants per cross bulked. After F_2_ generation, we adopted the single-seed descent method to advance F_2_ to F_4_ by generating 100 seeds per cross, which resulted in a total of 3000 F_4_ individuals. Selection was conducted at F_4_ by selecting a new set of 50 safflower individuals from the combination of the 3000 F_4_ and the 50 initial crossing parents. The 50 newly selected individuals were used as the crossing parents for cycle 2. We repeated the process up to cycle 5, and the genetic gains were calculated in four GS selection strategies for GY and OL at each cycle. The simulation procedure was repeated in 50 replicates, and we reported the averages.

### 2.2. Initial Phenotypes and Genotypes

The safflower diverse population sourced from the Australian Grain GenBank was used as the initial population in the study. The accession information, field design, and genotyping details of this diverse population have been described previously [30]. Briefly, the safflower collection was evaluated in four field trials, which were conducted in 2017 and 2018 with two trials each year in Victoria, Australia. All field trials adopted a randomized complete block design with 2–3 replications, and the plot size was 1 m × 5 m with five rows in each plot. Grain yield (GY) was determined as kg/plot and converted to t/ha with the width of 0.5 m between plots. Seed oil content (OL%) was determined by near-infrared reflectance spectroscopy (NIR) calibrated by the Soxhlet extraction method with an R-squared value of 0.95 and a standard error of prediction of 1.2%. We combined sites with a mixed linear model mentioned in [31] to estimate the best linear unbiased estimates (BLUEs) for GY (t/ha) and OL (%) for each individual. Narrow sense heritability (h^2^) was estimated by a genomic best linear unbiased prediction model (GBLUP) with BLUEs fitted as ‘phenotypes’, and it was 0.54 for GY and 0.8 for OL, and the genetic correlation between these two traits’ BLUEs was around 0.19. The accessed safflower diverse population were all genotyped using a genotyping-by-sequencing (GBS) assay. After removing SNPs with >50% missing and a minor allele frequency < 0.01, a total of 6911 SNPs remained, and missing genotypes were imputed by LinkImpute [32]. The genomic relationship matrix (GRM) was calculated according to VanRaden to reflect the relationship between accessions and breeding lines [33].

### 2.3. Simulation of the Proposed GS

Four GS selection strategies based on GEBVs or indices were compared. GS_GY and GS_OL are two single-trait selection strategies, and selection was based on the independent culling method by selecting the top 50 individuals with high GEBVs estimated by the single-trait GBLUP model, as detailed in [31]. In GS_GY + OL, a selection index was constructed to simultaneously select for GY and OL by combining both traits’ standardized GEBVs with equal weights, and the top individuals with the highest index were selected [31]. The GS_GY + OL + Rel strategy was similar to the GS_GY + OL, but the selected candidates had low relationships, which was achieved by simultaneously controlling crossing parent selection and mate allocation. In brief, a fitness matrix was generated by the mid-parent GEBVs of all candidature combinations minus the co-ancestry values of the corresponding pairs in GRM, and it was then fitted in the Genetic Algorithm (GA) to search for the optimized set of crosses (in our study, the set of 50 parents) with maximized fitness, as detailed in [16]. λ was used as a scalar with λ = 0.5 in the study to penalize genomic relationships. The 30 parents used for initial crosses in cycle one were randomly chosen, and a stochastic simulation in-house script was used to generate the crossing and the progeny genotypes with the recombination rate following a Poisson distribution.

GEBVs of safflower individuals were estimated in two steps. First, the allele effects (β) for each SNP were estimated by using BLUEs as phenotypes in BayesR [34], which assumes a mixture of four normal distributions N~(0, 0|0.0001|0.001|0.01) for GY and OL, respectively. Second, the GEBVs (ĝ) in the selection candidates were calculated by multiplying genotypes by the estimated marker effects:$$ĝ=X^{\prime }\beta$$
where *X*′ is the matrix of simulated genotypes (0, 1, and 2) for crossing progeny, and *β* is a vector of allele effects estimated for each trait.

### 2.4. Genetic Gain and Inbreeding

Genetic gain was the genetic improvement expressed in genetic standard deviation unit ($\sigma_{G}$), to be able to compare the gain in GY and OL:$${\Delta G}_{\;}=(\mathrm{mean}({GEBV}_{i})-\mathrm{mean}({GEBV}_{j}))/{\;\sigma }_{{GEBV}_{cycle\;1}}$$
where GEBV*_i_* and GEBV*_j_* are the average breeding values estimated for parents in cycles *i* and *j* (cycle *i* + 1), respectively. We used${\;\sigma }_{{GEBV}_{cycle\;1}}$ in the formula to be able to attain a fair comparison of genetic gains across cycles.

The inbreeding coefficient (F) was estimated as the mean of diagonal elements of GRM minus 1. Generally, low average F in the population indicates low inbreeding and high genetic diversity. The rates of inbreeding were calculated as [35]
ΔF*_ij_* = F*_j_* − F*_i_*,
where F*_i_* and F*_j_* are the mean inbreeding coefficients in cycles *i* and *j* (cycle *i* + 1), respectively.

The average genetic gain and inbreeding was plotted against each cycle for comparison.

## 3. Results

### 3.1. Initial Phenotypes and Genotypes

The distributions of BLUEs for GY and OL in the diverse safflower panel are shown in Figure 2a. The average BLUEs were around 2–4 t/ha for GY, and the highest yield exceeded 5.1 t/ha. OL ranged from 29 to 34% with few individuals exhibiting a percentage less than 25% and more than 40%. The highest OL was 42.4%. The GRM constructed from 6911 SNPs revealed that the 349 safflowers could be grouped into a few subgroups with a few safflower accessions or lines closely related to each other (Figure 2b). Selection based on the performance of the 349 safflower accessions or lines could be impacted by selecting the individuals within the same subgroups.

### 3.2. Genetic Gains

We randomly selected 50 out of 349 safflower accessions or lines to initiate the first crossing cycle and a total of five cycles were simulated. The mean GEBVs of each cycle for GY and OL are shown in Figure 3, which indicated a positive genetic improvement for target traits across cycles in different GS methods. The GS_GY selection strategy showed the GY, whereas GS_OL showed the highest genetic improvement for OL at all cycles. The single-trait strategies indicated the highest genetic improvement (mean GEBVs) for the target traits. After five cycles, the mean GEBVs for GY and OL improved substantially and were about 1 and 0.7 greater than the first cycle, respectively. However, the average GEBVs were poor for the trait not under selection in the single-trait GS method. With simultaneous selection of GY and OL, the GS_GY + OL showed slightly higher genetic improvement in both GY and OL at the early cycles (c2 and c3) compared to the GS_GY + OL + Rel strategy. However, the difference was completely diminished for GY and became negligible for OL in cycles four and five. The variation in GEBVs was large at the initial cross for both traits. As the cycle number increased, the variation decreased, but the rate of reduction in genetic variation differed between GS selection methods.

We calculated the genetic gain for each cycle (Table 1) and observed that the greatest gain was achieved at cycle one for all the GS methods. The gains were higher for the single-trait GS strategy than for multi-trait GS selection strategies. Within the single-trait GS methods, the gain for OL in the GS_OL method was higher than the gain for GY in GS_GY in the first cycle; however, the gain for GY was slightly greater after cycle one. After five cycles, single-trait selection resulted in gains of 2.609 and 2.777 in GY and OL, respectively, via two distinct breeding strategies. When simultaneously selecting for GY and OL, the GS_GY + OL strategy showed larger gain than did the GS_GY + OL + Rel in the first cycle. After five cycles, the sum of gains was close for the two multi-trait GS selection methods at about 1.8 for GY and 2.2 for OL.

### 3.3. Inbreeding Coefficient

The inbreeding coefficient (F) showed an increasing trend for all GS strategies (Figure 4). A sharp increase was seen from the initial cycle to cycle two, followed by a gradual increase in average F, indicating a huge loss in genetic diversity in the first breeding cycle. GS_GY had a higher inbreeding coefficient than GS_OL. Compared with single-trait GS strategies, GS_GY + OL and GS_GY + OL + Rel had lower inbreeding coefficients. GS_GY + OL + Rel had lower inbreeding than GS_GY + OL did at cycle two; however, the difference between the two multi-trait strategies was minimal after cycle two, which could indicate that the small population size makes inbreeding inevitable to a great extent.

## 4. Discussion

GS is a modern breeding tool used in plant breeding programs for germplasm improvement and variety development. The recurrent GS scheme can lead to a rapid increase in the frequency of favorable alleles in the breeding system to improve the germplasm and create new variations. Simulation of the recurrent GS scheme before practical implementation would allow the comparison of different GS strategies and facilitate optimization of the breeding program.

### 4.1. Genetic Gain and Inbreeding at the Early Cycle

Genetic gain is used to measure genetic improvement or genetic progress in breeding programs, and breeders are expected to keep long-term genetic gains within breeding programs to meet increasing demands. Factors affecting genetic gain include genetic variation within the breeding germplasm, selection intensity, accuracy of genetic predictions, and length of the breeding cycle [35]. In our study, we observed a sharp increase in genetic gains in the first GS selection cycle with all GS methods, which indicated that GS selection was effective, especially at the early breeding stage. This was in line with another recurrent selection simulation study, which demonstrated that GBLUP-based GS showed greater genetic gains than PS under the additive model, particularly in early selection cycles [36]. In a sorghum GS simulation study, a 12–88% gain advantage for traits controlled by major genes and a 26–165% gain increase for polygenic traits in the first few breeding cycles compared to conventional breeding methods were indicated [37].

The large standard deviation of the mean GEBVs in the first cycle indicated large genetic variation in the initial crossing parents, which could be the reason for the higher gain at the first cycle. The sharply dropped variation at cycle two, especially for single-trait GS strategies could be due to the selection of closely related individuals who carry QTL underlying GY and OL, as reported in a previous GWAS [38]. This assumption that the selected individuals were closely related was confirmed by the large increase in the average inbreeding coefficients calculated from the diagonal GRM after the first cycle. Hence, the genetic gain is achieved by reducing genetic variation within the selected population which makes continuous high genetic gain across cycles challenging [39]. Simultaneous selection for GY and OL in the GS_GY + OL strategy also showed the same trend that it had higher gain in the first cycle than GS_GY + OL + Rel did, but the gains and inbreeding coefficient after five cycles were quite close for the two methods. This indicated that controlling inbreeding in GS schemes to maintain long-term genetic gain is essential. Restricting parents’ level of co-ancestry to maximize ΔG by using GRM could effectively reduce average progeny inbreeding [40,41]. Our study confirmed that parental selection combined with mate allocation effectively reduced the inbreeding. Furthermore, the cost of reduced genetic gain while controlling for inbreeding in the first cycle could be compensated for by higher gain in subsequent breeding cycles [42].

### 4.2. Multi-Trait Genomic Selection

As a potential oil seed crop, safflower GY and OL are two major breeding targets in breeding programs [43,44]. Without the availability of economic values for these traits, we gave them equal economic importance in our simulation. The genetic correlation between the two, GY and OL, ranged from negative to low positive in different environments [31]. We considered a low positive genetic correlation of 0.19 between GY and OL, which resulted in a small favorable correlated response in the single-trait GS methods although negative correlations imply an unfavorable response in one trait when selecting for another trait. Compared to the expected gains based on the progenies resulting from the crossing of two independent culling breeding schemes, GS_GY and GS_OL, referred to as the reciprocal recurrent GS [45], the GS index GS_GY + OL method revealed higher gains for GY (1.782) and OL (2.243) than 1.564 for GY ((2.609 + 0.519)/2) and 1.401 for OL ((0.025 + 2.777)/2). Hence, we confirmed that index selection for multiple-trait selection was more efficient than independent culling [46]. Additionally, with mate allocation and parental selection (GS_GY + OL + Rel), multi-trait GS with indices could maintain a low inbreeding while achieving long-term genetic gain. This finding is consistent with a wheat grain yield and heat stress tolerance (HST) breeding study, which revealed that GS index selection with mating constraints resulted in long-term genetic gains in grain yield with adaptable HST whereas truncation selection caused a rapid loss in genetic diversity and a decrease in long-term genetic gains [47]. Our results demonstrated that aggregating favorable high-fatty-acid and high-yield alleles via the GS index method will potentially be the optimum strategy to develop new varieties which have both a high yield and a high oil content. However, index selection is less flexible for selecting certain primary traits while keeping other traits within a desirable range [48]. In safflower, breeding schemes which can maintain high genetic gain for yield but with increasing gain for the oil component need further study.

### 4.3. Optimization of the Breeding Program

When GS and PS are applied to the same breeding population, GS can achieve higher genetic gains by increasing selection accuracy in selection candidates and shortening the breeding interval. In the recurrent selection schemes, selection conducted in F_4_ is aimed at increasing the favorite allele frequencies to maximize homozygosity in the inbred lines, which could be used as parents in the next breeding cycle. However, with GS, selection could be conducted at the F_1_ generation to choose the best cross instead of the best lines [49]. Our simulation study showed that shortening recurrent GS breeding intervals could be achieved by applying early generation selection. Gaynor et al. compared a rapid recurrent selection scheme in which selection occurred at the F_1_ stage for population improvement via two-part breeding strategies and reported that 1.31 and 1.46 times greater genetic gains were achieved than under the standard GS recurrent selection [50]. In a lentil simulation study, GS selection was suggested for F_2_ instead of F_4_ to shorten cycle time, which could further increase genetic gains [51]. Furthermore, the greater gain achieved at the early breeding cycles in the recurrent scheme in our study indicated that shortening the cycle time could also be used to optimize the breeding program. Corjanc et al. [52] studied the impact of cycle numbers in a recurrent scheme on genetic gain and found that four cycles per year could achieve 15% higher long-term gains than truncation selection. A shortened cycle time combined with the shortest line fixation time was suggested to expedite the rapid recycling of parents in the breeding program through recurrent selection to enhance and accelerate the genetic improvement rate in developing irrigated rice [53]. Hence, to optimize the safflower breeding program, shorter recurrent GS breeding intervals could be achieved by applying early generation selection and fewer breeding cycles. In addition, shortening breeding intervals will dramatically improve the cost effectiveness of the breeding program, thus mitigating the challenges of limited resources and funding faced by safflower or other orphan crop breeding.

## 5. Conclusions

Safflower is an “orphan” crop and applying GS in its breeding program could help to fast-track the genetic diversity preserved in the genebank collection to facilitate rapid and efficient germplasm improvement and variety development. Using computer simulation to compare different GS strategies to optimize breeding programs, we found that a GS strategy with inbreeding control and simultaneous selection for GY and OL could achieve long-term genetic gains for both traits while decreasing the loss in genetic diversity in safflower. Early generation selection and shortened breeding cycles could further enhance genetic gains and maintain genetic diversity within breeding programs.

## Figures and Tables

**Figure 1 plants-13-01577-f001:**
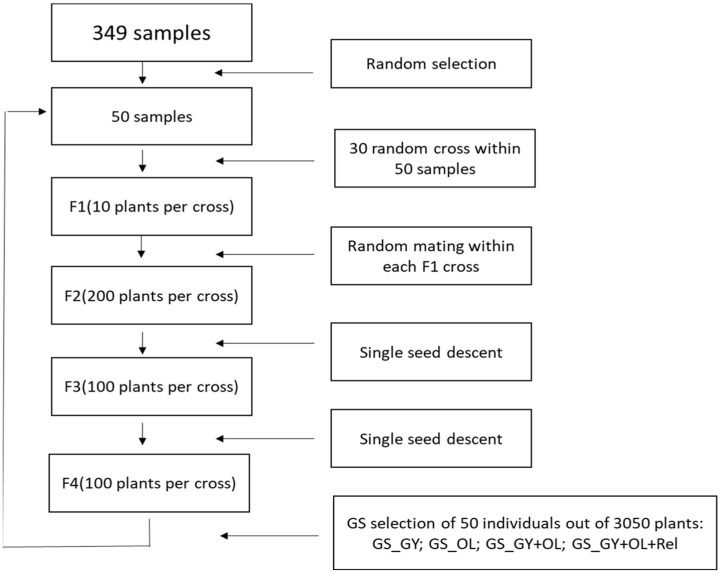
Workflow summary of the GS selection procedures used in the simulation for the development of selection cycles.

**Figure 2 plants-13-01577-f002:**
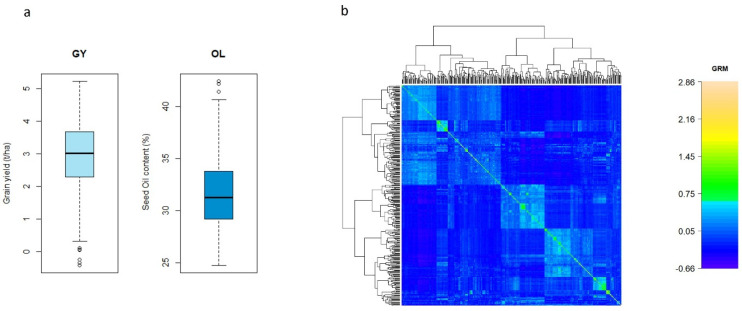
(**a**) Boxplot combined BLUEs for grain yield (GY) and seed oil content (OL) in 349 diverse safflower accessions. (**b**) Heatmap of genomic relationships based on genomic relationship matrix (GRM), where higher values indicate higher relatedness.

**Figure 3 plants-13-01577-f003:**
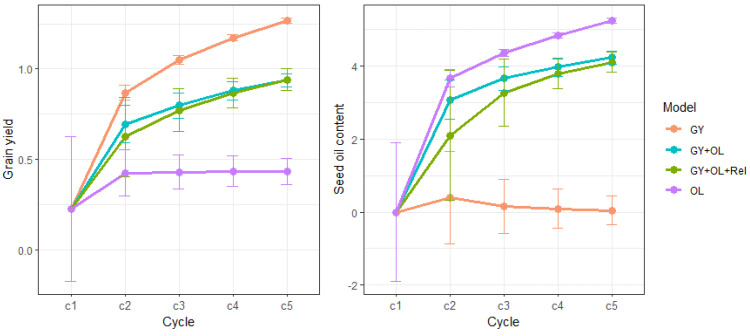
Mean and standard deviation of GEBVs in four GS selection strategies across simulated breeding cycles for grain yield (**left**) and seed oil content (**right**).

**Figure 4 plants-13-01577-f004:**
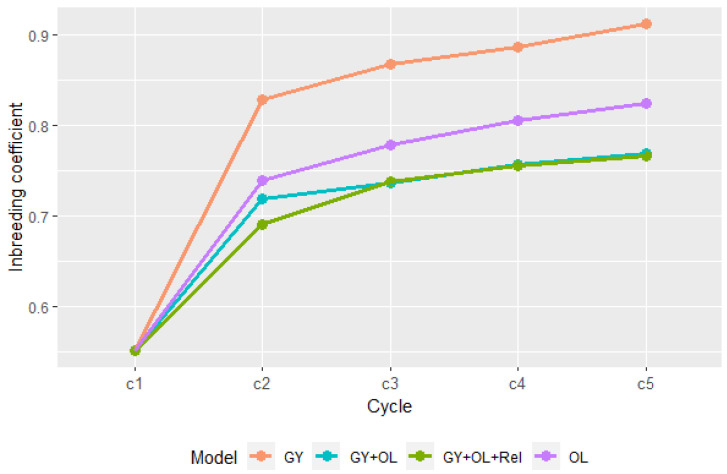
The inbreeding coefficient of four GS selection strategies at different cycles in the simulation.

**Table 1 plants-13-01577-t001:** Genetic gains for grain yield and oil content at different cycles of four selection strategies.

Trait	Cycles	GS Model			
		GY	OL	GY + OL	GY + OL + Rel
GY	c1	1.609	0.499	1.175	0.995
	c2	0.454	0.013	0.256	0.373
	c3	0.308	0.01	0.206	0.233
	c4	0.238	−0.003	0.145	0.188
	Sum	2.609	0.519	1.782	1.789
OL	c1	0.204	1.939	1.632	1.112
	c2	−0.125	0.365	0.305	0.618
	c3	−0.029	0.254	0.166	0.279
	c4	−0.025	0.219	0.140	0.163
	Sum	0.025	2.777	2.243	2.172

## Data Availability

The phenotypic dataset and the genotype dataset supporting the conclusions of this article can be found at doi:10.3389/fgene.2023.1129433.

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
