# Peer review of "Genetic Gain and Inbreeding in Different Simulated Genomic Selection Schemes for Grain Yield and Oil Content in Safflower"

_plants, 2024, doi:10.3390/plants13111577_

Round 1

Reviewer 1 Report

Comments and Suggestions for Authors

In this study, the authors simulated four GS strategies to compare genetic gains and inbreeding rates during breeding cycles in a safflower recurrent selection breeding program targeting grain yield (GY) and seed oil content (OL).

In result section, they observed positive genetic gains over cycles in all four GS strategies, where the first cycle delivered the largest genetic gain. Single traits GS strategies had the greatest gain for the target trait but with very limited genetic improvement for the other trait. Simultaneous selection for GY and OL via index indicated higher gains for both traits than crossing between the two single trait independent culling strategies. The multi-trait GS strategy with inbreeding control (GS_GY+OL+Rel) showed lower inbreeding rate but similar gain compared to GS_GY+OL (without inbreeding control) after few cycles. Although this study has achieved certain results, most results were descriptive. Meanwhile, I think the workload did not meet the requirements of Plants magazine.

In discussion section, two safflower genomes have been reported in 2021 and 2023, the authors should discuss the roles of whole genome sequencing in GS besides genotyping-by-sequencing mentioned study.

Therefore, at the current state, I do not recommend publishing this study in this journal.

Minor comments:

Line 20, the grammatic errors, such as “The the multi-trati”.

Line 28, the full name of GEBVs should be added for the first time.

Comments on the Quality of English Language

Quality of English Language is good.

Author Response

Answer to the major comments:

1) In this study, the authors simulated four GS strategies to compare genetic gains and inbreeding rates during breeding cycles in a safflower recurrent selection breeding program targeting grain yield (GY) and seed oil content (OL).

In result section, they observed positive genetic gains over cycles in all four GS strategies, where the first cycle delivered the largest genetic gain. Single traits GS strategies had the greatest gain for the target trait but with very limited genetic improvement for the other trait. Simultaneous selection for GY and OL via index indicated higher gains for both traits than crossing between the two single trait independent culling strategies. The multi-trait GS strategy with inbreeding control (GS_GY+OL+Rel) showed lower inbreeding rate but similar gain compared to GS_GY+OL (without inbreeding control) after few cycles. Although this study has achieved certain results, most results were descriptive. Meanwhile, I think the workload did not meet the requirements of Plants magazine.

Response: Simulation breeding program has its long history; however, it has never been done in safflower breeding. In this article, we applied computer simulation in a recurrent GS breeding scheme to generate fundamental information which can be applied in a practical breeding program. 

2) In discussion section, two safflower genomes have been reported in 2021 and 2023, the authors should discuss the roles of whole genome sequencing in GS besides genotyping-by-sequencing mentioned study.

Response: The article was focused on applying the quantitative genetic theory and knowledge generated by research in other crops in safflower, and the comparison is the genetic gain and inbreeding rate during the breeding cycles. 

3) Therefore, at the current state, I do not recommend publishing this study in this journal. 

Response: New published safflower assembly are the important genomic resources for safflower, and they could dramatically improve the accuracy of SNPs identification and facilitate SNP annotation in GWAS. In GS, it would benefit the prediction accuracy. But the paper's focus wasn't the prediction accuracy. 

Minor comments:

Line 20, the grammatic errors, such as “The the multi-trati”.

Line 28, the full name of GEBVs should be added for the first time.

Response: Edited as suggested. Please see the re-submit draft.

Reviewer 2 Report

Comments and Suggestions for Authors

The following modifications are to be made to the manuscript:

(1) Safflower being an Oilseed crop, the use of 'seed yield' would be more appropriate than 'grain yield'. The authors need to revise throughout the manuscript.

(2) Line 20: Remove the duplicate 'the'

(3) Line 82: Correct the spelling of 'oil'

(4) Figure 1: workflow: Correct the spelling of 'single seed descent'

(5) Line 113: Revise as 'Initial phenotyping and genotyping'

(6) Results to be elaborated further

(7) Elaborate the discussion by citing from the other crops where significant work has been carried out on multi-trait genomic selection.

Author Response

Response to the comments:

The following modifications are to be made to the manuscript:

(1) Safflower being an Oilseed crop, the use of 'seed yield' would be more appropriate than 'grain yield'. The authors need to revise throughout the manuscript.

response:  grain yield is popular used name for yield, especially the abbreviation of GY was easy to understand. Therefore, to avoid confusion of the audience, I would like to use grain yield in the manuscript. 

(2) Line 20: Remove the duplicate 'the'

response: revised

(3) Line 82: Correct the spelling of 'oil'

 response:  revised

(4) Figure 1: workflow: Correct the spelling of 'single seed descent'

4) revised

(5) Line 113: Revise as 'Initial phenotyping and genotyping'

response:   revised

(6) Results to be elaborated further

response:  revised

(7) Elaborate the discussion by citing from the other crops where significant work has been carried out on multi-trait genomic selection.

response:  Discussion have been revised. Multi-trait GS was cited for optimizing breeding programs in ryegrass, lentil, wheat, rice etc. 

Reviewer 3 Report

Comments and Suggestions for Authors

This study simulated four GS strategies to compare genetic gains and inbreeding rates during breeding cycles in a safflower recurrent selection breeding program targeting grain yield (GY) and seed oil content (OL). The description and representation are well. However they are many minor points should be revised.

L21. Definition ‘Rel’.

L68. testedfor?

L75. ‘and [23]’?

L94. Fig1; L108. F1; L167. Fig. 1a. L210. ‘Figure 3’.  They are confused.

L103. ‘ofthe'?

Author Response

Response to the reviewer:

1)  This study simulated four GS strategies to compare genetic gains and inbreeding rates during breeding cycles in a safflower recurrent selection breeding program targeting grain yield (GY) and seed oil content (OL). The description and representation are well. However they are many minor points should be revised.

L21. Definition ‘Rel’.

Response: revised

2)  L68. testedfor?

Response:  revised

3)  L75. ‘and [23]’?

Response:  revised

4)  L94. Fig1; L108. F1; L167. Fig. 1a. L210. ‘Figure 3’.  They are confused.

Response:  re-ordered the figures

5)  L103. ‘ofthe'?

Response:  revised 

Reviewer 4 Report

Comments and Suggestions for Authors

There are a few suggestions that can be made to improve the  introduction:

1. The authors could provide a brief explanation of how GS differs from traditional phenotypic selection (PS) and marker-assisted selection (MAS) to help readers understand the advantages of GS more clearly.

2. The authors mention the use of simulation in plant breeding but could provide more context on why simulation is particularly useful for comparing GS strategies and optimizing breeding programs.

There are a few areas where additional information or clarification could be provided to improve the understanding of the study:

1. Simulation Outline:

   - Provide more details on the specific parameters used in the simulation, such as the population size, number of crosses, and number of generations.

   - Clarify how the initial 50 individuals were selected from the diverse safflower accessions and breeding lines.

2. Initial Phenotypes and Genotypes:

   - Provide more information on the field trial design, such as the number of replications, plot sizes, and locations.

   - Explain how the narrow-sense heritability values for GY and OL were estimated.

3. Simulation of the Proposed GS:

   - Clarify how the weights for GY and OL were determined in the selection index for the GS_GY+OL strategy.

4. Genetic Gain and Inbreeding:

   - Provide the equations used for calculating genetic gain and inbreeding rate to improve clarity.

5. Statistical Analysis:

   - Include a section on the statistical analyses performed, such as the methods used for comparing genetic gains and inbreeding rates across GS strategies and cycles.

There are a few areas where the authors could improve the results:

1. Initial Phenotypes and Genotypes:

   - Discuss the implications of the observed genetic relationships among the safflower accessions and breeding lines for the simulation study.

2. Genetic Gains:

   - Consider presenting the genetic gains in table 1 in addition to the figures.

   - Discuss the statistical significance of the differences in genetic gains between GS strategies and cycles.

3. Inbreeding Coefficients:

   - Discuss the statistical significance of the differences in inbreeding coefficients between GS strategies and cycles.

There are a few areas where the authors could expand the discussion:

1. Genetic Gain and Inbreeding at the Early Cycle:

   - Discuss the potential factors contributing to the large genetic gains observed in the first cycle, such as the high genetic diversity in the initial population and the effectiveness of GS in capturing both major and minor effect loci.

2. Multi-Trait Genomic Selection:

   - Compare the findings of this study with previous studies on multi-trait GS in other crops and discuss any common themes or unique aspects of safflower breeding.

3. Optimizing the Breeding Program:

   - Discuss the potential benefits and challenges of implementing early generation selection and shortening breeding cycles in the context of safflower breeding, considering factors such as genotyping costs, phenotyping resources, and the rate of genetic progress.

4. Implications:

   - Discuss the potential implications of the findings of this study for the wider adoption of GS in minor crops or orphan crops with limited resources and genetic diversity.

Author Response

Respond to the reviewer:

Introduction:

  1. The authors could provide a brief explanation of how GS differs from traditional phenotypic selection (PS) and marker-assisted selection (MAS) to help readers understand the advantages of GS more clearly.

Author: The first paragraph is about what the GS is. The second paragraph is about the GS could achieve higher genetic gain than PS or MAS, showed in line 34 -40 and examples followed. I re-framed the sentence to make it clearer for readers. 

  1. The authors mention the use of simulation in plant breeding but could provide more context on why simulation is particularly useful for comparing GS strategies and optimizing breeding programs.

Author: Lin 67-71 mentioned simulation is useful and cost-effective tool for comparing GS strategies. Extra sentence was added in line 68-69.

There are a few areas where additional information or clarification could be provided to improve the understanding of the study:

  1. Simulation Outline:

   - Provide more details on the specific parameters used in the simulation, such as the population size, number of crosses, and number of generations.

   - Clarify how the initial 50 individuals were selected from the diverse safflower accessions and breeding lines.

Author: line 102-112 have detailed about the design of the simulation, including the information reviewer asked.

  1. Initial Phenotypes and Genotypes:

   - Provide more information on the field trial design, such as the number of replications, plot sizes, and locations.

   - Explain how the narrow-sense heritability values for GY and OL were estimated.

Author: The field trial info was added in line 123-128. and narrow-sense heritability estimation was added at line 131-133.

  1. Simulation of the Proposed GS:

   - Clarify how the weights for GY and OL were determined in the selection index for the GS_GY+OL strategy.

Author: line 146-148 describe how the weights was developed.

  1. Genetic Gain and Inbreeding:

   - Provide the equations used for calculating genetic gain and inbreeding rate to improve clarity.

Author: line 171 and line 178 are the equations used.

  1. Statistical Analysis:

   - Include a section on the statistical analyses performed, such as the methods used for comparing genetic gains and inbreeding rates across GS strategies and cycles.

Author: Genetic gain at each cycle was calculated as the average of the GEBVs with the standard deviation (50 reps).  line 181 was added to indicate the comparison. 

There are a few areas where the authors could improve the results:

  1. Initial Phenotypes and Genotypes:

   - Discuss the implications of the observed genetic relationships among the safflower accessions and breeding lines for the simulation study.

Author: line 194-195 was added. 

  1. Genetic Gains:

   - Consider presenting the genetic gains in table 1 in addition to the figures.

   - Discuss the statistical significance of the differences in genetic gains between GS strategies and cycles.

Author: Fig. 2. was used to visualize the results. 

The comparison was made in terms of genetic gain and inbreeding coefficient. No multiple comparison was conducted for this case.

  1. Inbreeding Coefficients:

   - Discuss the statistical significance of the differences in inbreeding coefficients between GS strategies and cycles.

Author: The comparison was conducted by the inbreeding coefficient. No multiple comparison was conducted in this case. Reference:  https://doi.org/10.1371/journal.pone.0235554

There are a few areas where the authors could expand the discussion:

  1. Genetic Gain and Inbreeding at the Early Cycle:

   - Discuss the potential factors contributing to the large genetic gains observed in the first cycle, such as the high genetic diversity in the initial population and the effectiveness of GS in capturing both major and minor effect loci.

Author: Please see line 266-280 is about the discussion as reviewer suggested.

  1. Multi-Trait Genomic Selection:

   - Compare the findings of this study with previous studies on multi-trait GS in other crops and discuss any common themes or unique aspects of safflower breeding.

Author: For multi-trait GS, the focus is which breeding strategy is more effective, independent culling or selection index. Because we only compared about genetic gain for GY and OL, we suggested using selection index to simultaneously breeding for both traits. See line 301-320.

  1. Optimizing the Breeding Program:

   - Discuss the potential benefits and challenges of implementing early generation selection and shortening breeding cycles in the context of safflower breeding, considering factors such as genotyping costs, phenotyping resources, and the rate of genetic progress.

Author:  line 346-349 was added to discuss the potential benefits of shortening breeding intervals.

  1. Implications:

   - Discuss the potential implications of the findings of this study for the wider adoption of GS in minor crops or orphan crops with limited resources and genetic diversity.

Author; Line 346-349 added for the potential benefits to the orphan crops  without stretching too much. 

Reviewer 5 Report

Comments and Suggestions for Authors

-          This paper correspond for scope of journal.

-          The title corresponds to the content of the paper. 

-           

-          Research topic is original and relevant in field of investigation.

-          This study have significant contribution for improving breeding methods of safflower on the base of using  simulated four genomic selection (GS) strategies and comparison of genetic identified gains and inbreeding rates during breeding cycles in a safflower recurrent selection breeding program targeting grain yield and seed oil content. 

-          This study addressed to establish  and implement genomic selection (GS) in breeding program to take adantage of this modern breeding tool to breed safflower in the aim to  increasing both grain yield and oil content in safflower.

-          The aim of study should be particular paragraph (last paragraph on the end of chapter of Introduction) and clearly and precisely pointed out. The goal of the research should be written explicitly. You can achieve this by reconstructing the text from line 85 to 89.

For example: The aim of work was study of (i) simulated a safflower recurrent breeding program targeting grain yield (GI) and seed oil content (OL) (ii) comparison of the genetic gains and losses of genetic diversity during each breeding cycle and (iii) establish advantage of genomic selection  for improvement of those two traits in comparison with phenotypic selection.

-          Key words are appropriate.

-          Scientific methodology is appropriate.

-          Results are clearly presented and discussed.

-          Tables, figures, pictures are clear.

-          The conclusions are written clear and based on research results

-          Manuscript is acceptable after minor corrections!

-          Remarks: It should be write title of Figure 5

-          Sugestion: On line 103 two words are joined “…. ofthe  ….”, there should be a space between them  “…of them…”

Author Response

Respond to reviewer comments:

1) the aim of the study had been re-formatted as suggested in line 93-95;

2) Figure tile had been checked and added.